# Chemosensory Receptor Expression in the Abdomen Tip of the Female Codling Moth, *Cydia pomonella* L. (Lepidoptera: Tortricidae)

**DOI:** 10.3390/insects14120948

**Published:** 2023-12-14

**Authors:** William B. Walker III, Alberto M. Cattaneo, Jennifer L. Stout, MacKenzie L. Evans, Stephen F. Garczynski

**Affiliations:** 1Temperate Tree Fruit and Vegetable Research Unit, United States Department of Agriculture—Agricultural Research Service, Wapato, WA 98951, USA; jennifer.stout@usda.gov (J.L.S.); mackenzie.evans@usda.gov (M.L.E.); 2Chemical Ecology Group, Department of Plant Protection Biology, Swedish University of Agricultural Sciences, Lomma (Campus Alnarp), 234 56 Skåne, Sweden; albertomaria.cattaneo@slu.se

**Keywords:** odorant receptors, gustatory receptors, ionotropic receptors, codling moth, RNA-seq, gene expression, transcriptome, ovipositor

## Abstract

**Simple Summary:**

The codling moth, *Cydia pomonella,* is a major global agricultural pest of apple, pear, and walnut, and its behaviors are largely influenced by taste and smell. Different protein families that function as chemosensory receptors mediate detection of compounds from the environment. Manipulation of codling moth behaviors by using these compounds in orchards has been a successful approach to reduce fruit damage. This approach, however, has largely targeted male behaviors. Since it is the females that lay eggs, which propagate the species, in this report we sought to learn more about female behaviors. Specifically, we examined gene expression of the smell and taste receptors in female codling moth abdomen tip organs involved in egg-laying behaviors. We identified sets of both smell and taste receptor genes expressed in the abdomen tip that are likely candidates to influence egg-laying decisions. We also determined that expression of some of these receptors is increased or decreased after mating. These results increase our knowledge about the genes that can influence female codling moth behaviors. Future research will study the role of individual receptors with the aim of identifying taste and smell compounds that can be used to manipulate female codling moth behaviors in orchards.

**Abstract:**

In insects, the chemical senses influence most vital behaviors, including mate seeking and egg laying; these sensory modalities are predominantly governed by odorant receptors (ORs), ionotropic receptors (IRs), and gustatory receptors (GRs). The codling moth, *Cydia pomonella*, is a global pest of apple, pear, and walnut, and semiochemically based management strategies limit the economic impacts of this species. The previous report of expression of a candidate pheromone-responsive OR in female codling moth ovipositor and pheromone glands raises further questions about the chemosensory capacity of these organs. With an RNA-sequencing approach, we examined chemoreceptors’ expression in the female codling moth abdomen tip, sampling tissues from mated and unmated females and pupae. We report 37 ORs, 22 GRs, and 18 IRs expressed in our transcriptome showing overlap with receptors expressed in adult antennae as well as non-antennal candidate receptors. A quantitative PCR approach was also taken to assess the effect of mating on OR expression in adult female moths, revealing a few genes to be upregulated or downregulating after mating. These results provide a better understanding of the chemosensory role of codling moth female abdomen tip organs in female-specific behaviors. Future research will determine the function of specific receptors to augment current semiochemical-based strategies for codling moth management.

## 1. Introduction

The chemical senses, olfaction and gustation, underlie most vital behaviors in insects including food- and mate-seeking, feeding, egg-laying (oviposition), and predator avoidance. The olfactory system of insects is primarily located in appendages on the head, with the antenna serving as the primary sensory organ; likewise, olfactory function is also found in the maxillary and labial palps [1]. These organs are especially essential for influencing behaviors at a distance, such as male moth orientation and attraction to females through olfactory detection of miniscule amounts of emitted pheromone molecules. Gustation is known to be more distributed, with taste function primarily occurring on the mouthparts and legs, among other appendages.

Recently, chemosensory function on the female ovipositor has come into focus, suggesting a direct role for detection of chemical cues by the ovipositor in precisely mediating oviposition site selection. Typically, in female moths, the ovipositor is anatomically close to the reproductive organs and sex pheromone gland, and it has thus been speculated that olfactory function in this region, at the distal abdomen tip, may influence oviposition, mating, and pheromone production behaviors. This pheromone gland ovipositor (PG-OV) complex, as it is commonly referred to in the literature, has also been observed in the distal segments of female codling moth, *Cydia pomonella,* abdomen tissue [2], and it was previously demonstrated that apple odorants influence oviposition behavior in the codling moth [3].

In insects, chemosensory detection is initiated by olfactory or gustatory sensory neurons (OSNs and GSNs), which project their dendrites into small porous hairs, known as sensilla, that cover the surface of the antennae and other chemosensory tissues. Chemosensory receptor proteins, which reside in the cellular membrane of OSN dendrites, are the molecular determinants responsible for directly interacting with odorant molecules; ligand binding by the receptor triggers conformational change of the receptor, ion flux, and ultimately neuronal membrane depolarization whereby the chemical signal of the odorant molecule is converted to an electrical message that is sent to the brain [4]. Two primary families of olfactory receptors have been identified in insects, first, the odorant receptor (OR) family [5,6,7], which within Lepidoptera includes well-defined clades of candidate pheromone receptors (PRs) [8,9], and subsequently, the ionotropic receptor (IR) antennal subfamily [10]. Similarly, taste receptors are found in the gustatory receptor (GR) family, which is evolutionarily related to the ORs, forming the chemosensory receptor superfamily [11,12]. Since the initial discovery of chemosensory receptors in the genetic model, *Drosophila melanogaster*, the advent of robust genomic and transcriptomic sequencing methodologies has yielded the identification and description of olfactory and gustatory receptor genes in numerous insect species [13], including within Lepidoptera [14].

A chemosensory role for insect ovipositors was hypothesized from morphological studies that identified multiporous sensilla, which are characteristic of olfactory function, on the surface of ovipositors in lepidopteran species [15,16]. Olfactory function of insect ovipositors has been confirmed with electrophysiological experiments. Single sensillum recordings from ovipositor sensilla showed clear neuronal responses to odorants in the lepidopteran species *Manduca sexta* [17] and *Helicoverpa assulta* [18]. Likewise, gustatory function has also been reported for the ovipositor of *S. littoralis*, based on morphological and physiological studies [19].

At the molecular level, initial reports identified few chemosensory receptors expressed in lepidopteran abdomen tips associated with PG-OV tissues [17,20,21]. In the first report on ORs expressed in female PG-OV, two PRs were identified through PCR and in situ hybridization assay in *Heliothis virescens*, and it was hypothesized that these receptors may play a role in the detection and feedback regulation of sex pheromone production [22]. Utilizing RNA-sequencing and transcriptomic analyses, some ORs plus the odorant receptor co-receptor (Orco), which functions as an OR trafficking chaperone and ion channel [23,24] were identified in ovipositor tissue of *Sesamia nonagrioides* [20]; two ORs, but not Orco, were identified in PG-OV tissue of *Chilo suppressalis* [21]; two ORs plus Orco, as well as two GRs and nine IRs, were identified in ovipositor tissue of *M. sexta* [17]. More recently, 12 ORs, 4 GRs, and 10 IRs have been found expressed in the PG-OV tissues of *Spodoptera littoralis*, including PRs, Orco, CO_2_ receptors, sugar receptors, and IR co-receptors [25], and a complex pattern of all these chemosensory receptors, along with other gene families known to function in chemosensory processes, was found expressed in the PG-OV of *Helicoverpa zea* [26]. In another report, the PG-OV tissue of *H. assulta* unveiled 22 ORs, 6 GRs, and 13 IRs [18]. In this report, a single OR, HassOR31, was functionally characterized and determined to detect a variety of known host plant odorants, and a role for some of these odorants in oviposition behavior mediated through the ovipositor was described [18].

In the codling moth, *C. pomonella*, the molecular determinants of chemosensory detection have been identified and characterized for the antennae of male and female codling moth adults and also in neonate larval heads [27]. As a frame of reference, 85 ORs, 65 GRs, and 39 IRs have been reported from the first characterized sequenced genome of *C. pomonella* [28]. A total of 58–66 ORs [27,29], 18 GRs, and 21 IRs [27] have been reported to be expressed in *C. pomonella* adult antennae. Notably, several ORs displayed either male or female antennae expression bias or specificity [27,29]; it is hypothesized that ORs that display sex-biased or specific expression may detect odorants involved in mediation of sex-specific behaviors.

To date, few codling moth ORs have been functionally characterized [30]. The candidate PR CpomOR3 was first identified to detect the plant volatile kairomone, pear ester [31,32], and subsequently found to respond to a lesser degree to the primary pheromone of *C. pomonella,* codlemone, (E,E)-8,10-dodecadienol [28]. Multiple receptors have been identified that detect the codlemone behavioral modulator odorant, codlemone acetate [29,32]. CpomOR19 was reported to detect several structurally related indanone volatiles of unknown behavioral and ecological significance [33]. Finally, the odorant receptor co-receptor, CpomOrco, demonstrated altered susceptibility to acidic pH and altered agonist binding when mutagenized in key residues of the intracellular loop 3 [34].

Until now, prior research on the molecular mechanisms and genes involved in chemosensation in *C. pomonella* has largely been focused on the antennae [27,29]. However, a study involving CRISPR editing of the candidate PR gene that encodes CpomOR1 [35] revealed that, in addition to male enriched antennal expression [27], CpomOR1 is also expressed in female abdomen tip [35]. Based on this finding and reports from other lepidopterans [17,18,20,21], it was hypothesized that other chemosensory receptor genes would be expressed in the *C. pomonella* abdomen tip. Due to the role of the female moth PG-OV complex in both mating and oviposition behaviors, RNA sequencing was conducted on *C. pomonella* abdomen-tip tissue derived from both mated and unmated female adults, as well as pre-adult pupae. For all chemosensory receptors identified in the assembled transcriptome, reverse transcription PCR was conducted to confirm the presence transcripts identified in the transcriptome. Due to hypothesized function of olfaction and ORs in mating behaviors, all candidate PRs from both the canonical [9] and recently reported novel PR clades [8] were examined in this manner, regardless of their presence or absence in the assembled abdomen transcriptome. Finally, it is hypothesized that mating may affect gene expression of ORs in female abdomen-tip tissues, with potential downstream consequences for olfactory-mediated behaviors. We thus conducted a qPCR study on a subset of ORs expressed in the female abdomen tip to assess whether transcriptional regulation factors into OR expression patterns in unmated versus mated adult female *C. pomonella*. Identification of chemosensory receptor repertoires in the female codling moth abdomen tip, including those for which gene expression is influenced by mating, will guide future research on elucidating the genetic mechanisms underlying female-specific behaviors of *C. pomonella*.

## 2. Materials and Methods

### 2.1. Insects and Sample Collection

A codling moth colony started from field-collected insects is maintained at the USDA-ARS in Wapato, WA. Larvae resulting from colony adults mating are reared on an artificial diet (Southland Products, Lake Village, AR, USA) at 26C, 50% RH, with a photoperiod of 16:8 (L:D) h. Under all experimental conditions, with reference to sampling for both RNA sequencing and qRT-PCR assay, adult females were individually pair-mated to one male in plastic resealable zipper storage bags within 2–3 days of eclosion from the pupal stage. Mating was confirmed after 2–3 days through visual inspection of successful egg laying, and mated females were thus collected for sampling. For experimental virgin females, age-matched controls were selected and handled in the same manner, except that they were not pair-mated in the plastic bags.

For female abdomen tip dissections, #5 Dumont biology grade tweezers (Ted Pella, Inc., Redding, CA, USA) were used to gently squeeze the female abdomen above segment 4 and the tip was removed by cutting at segment 2 with dissecting scissors over a 1.5-mL microfuge tube containing 500 µL of RNAlater (Ambion/Life Technologies, Carlsbad, CA, USA). For RNA sequencing, 20 to 30 unmated, mated, and pupal insects were sampled. For RT-PCR and qRT-PCR experiments, 20 to 30 abdomen tips from unmated or mated females were dissected for each sample. For PCR amplification controls, antennae from 50 males and 50 females were sampled using #5 Dumont biology grade stainless steel tweezers to pluck antennae at their base, and all antennae were combined into one 1.5-mL microfuge tube containing 0.5 mL of RNAlater.

### 2.2. RNA Sequencing

RNA was extracted from one each of the unmated, mated, and pupal female abdomen tip samples using the RNeasy Plus Mini Kit (Qiagen, Venlo, The Netherlands) according to the manufacturer’s protocol for animal tissues. Total RNA was eluted from the spin column with 40 µL of RNase-free water and assayed for quality and concentration with a Nanodrop 1000 spectrophotometer (Thermo Scientific, Waltham, MA, USA). Total RNA was shipped to Novogene Co., Inc (Chula Vista, CA, USA) for quality control, library preparation, and Illumina sequencing. RNA integrity was measured using the RNA Nano 6000 Assay Kit with the Agilent Bioanalyzer 2100 system (Agilent Technologies, Santa Clara, CA, USA). After quality control, mRNA was enriched using oligo(dT) beads, then fragmented randomly using fragmentation buffer. cDNA was synthesized from mRNA and random hexamer primers for first-strand synthesis, followed by second-strand synthesis with a custom Illumina second-strand synthesis buffer, dNTPs, RNaseH, and DNA polymerase I. Finally, after terminal repair and sequencing adaptor, ligation steps were performed, the double-stranded cDNA library was size selected and PCR enriched in preparation for sequencing. Samples were sequenced on an Illumina Hi-Seq 2000 platform with a paired-end (PE150) approach to generate raw sequence files for each sample in FASTQ format [36].

### 2.3. Transcriptome Assembly and Chemosensory Receptor Analyses

Low-quality raw sequence reads were filtered out by Novogene to generate clean-read FASTQ files by removing reads that contained adapters, contained greater than 10% ambiguous nucleotides, or contained low quality nucleotides (less than or equal to Q score of 5) for greater than 50% of all nucleotides. Trimmomatic software (version 0.32) was utilized to trim low-quality bases from the 3′ end of each read, with the TRAILING:20 command [37]. A single de novo transcriptome comprised of trimmed sequenced libraries from female pupal and adult mated and unmated abdomen tip samples was assembled using Trinity version v.2.8.4 [38], using default parameters, with a single Trinity.fasta file as output. To assess the completeness the female abdomen-tip transcriptome, an Arthropoda BUSCO database, consisting of 1013 core genes that are highly conserved single-copy orthologues, was used to query the transcriptomes. For this process, the gVolante web server, version 2.0.0 (https://gvolante.riken.jp/, accessed on 12 December 2023) was utilized with the following parameters: cut-off length for sequence statistics and composition: 1; sequence type: trans; selected program: BUSCO_v5; selected ortholog set: Arthropoda [39]. To facilitate identification of complete chemosensory receptor open reading frames (ORFs) to the greatest extent possible, a secondary transcriptome was generated and analyzed using the trimmed abdomen tip reads as well as the adult antennal reads from our previous report [27]. This secondary transcriptome was built using exactly the same approach as described above for our abdomen-tip-only samples.

For identification of chemosensory receptor genes expressed in female abdomen-tip tissues, tblastn searches were conducted against the transcriptome. For this, using command line, a blast nucleotide database was generated from the Trinity.fasta file and queried by protein sequence fasta files containing previously annotated *C. pomonella* OR, GR, and IR protein sequences [27]. Blast version 2.2.24+ was used to perform a tblastn query and a minimum e-score threshold of 1 × 10^−5^ was required for hits; additional parameters included num_descriptions 50 and blast output files were generated with output format six [40], with the following descriptors included: qlen qseqid slen sseqid stitle evalue bitscore score pident nident ppos positive sframe. For each of the previously annotated chemosensory receptor sequences, all blast hit transcript clusters were extracted from the Trinity.fasta file with an in-house command line script. Nucleotide sequences were translated into protein sequence with the ExPASy web Translate tool [41], and the protein sequences were aligned to reference annotations with the ClustalOMEGA web tool (http://www.ebi.ac.uk/Tools/msa/clustalo/, accessed on 12 December 2023) [42].

All OR, GR, and IR gene transcripts that did not correlate to previously annotated sequences [27] as well as those with incomplete ORFs were subsequently queried against the antenna-plus-abdomen-tip transcriptome as well as a *C. pomonella* genome database (http://v2.insect-genome.com/, accessed on 12 December 2023) [28] with a tblastn or blastp search, respectively, to identify and extract the most complete transcript sequence to the greatest extent possible for each expressed gene.

### 2.4. RT-PCR, Molecular Cloning and Sequence Analysis

Considering that our transcriptome was generated from non-replicated samples, and also our ultimate aim of semiochemical-based control of codling moth, we thus sought to confirm adult expression for all chemosensory receptors identified in the transcriptome. Due to the hypothesized role of olfactory sense in mating behaviors, we also assayed all candidate PRs whether or not they were identified in the abdomen-tip transcriptome, based upon their phylogenetic clustering in known Lepidoptera PR clades [8,9]. Using transcriptomic [27] or genomic [28] sequence information, PCR primers were designed using the IDT OligoAnalyzer tool (Integrated DNA Technologies, Coralville, IA, USA; https://www.idtdna.com/calc/analyzer, accessed on 12 December 2023), which may also be suitable for qRT-PCR asssay (Appendix A). Considering that not all primer sets work equally well, multiple primer sets were designed for each target chemosensory receptor gene. cDNA was generated with input of 1 μg of total RNA each from abdomen tip samples of mated and unmated codling moth females using the QuantiTech Reverse Transcription Kit cDNA synthesis kit (Qiagen, Venlo, The Netherlands) according to the manufacturer’s protocol. PCR assays were performed with the Dream Taq master mix system (Thermo Fisher Scientific, Waltham, MA, USA) on cDNA from single biological samples distinct from those used for RNA sequencing. Specific primer pairs were used for each chemosensory receptor gene and a sample containing cDNA derived from mixed male and female *C. pomonella* antennae was used as a positive control. For all PCR assays, thermocycling conditions were used for 40 cycles of 30s at 95 °C, 30s at 55–60 °C (depending upon primer annealing temperature), and 30s at 72 °C, with an initial 3 min initiation at 95 °C and 7 min final extension at 72 °C. PCR reactions were loaded on 1.5% agarose gels loaded with Gel Red stain (Biotium Inc., Fremont, CA, USA), and after electrophoresis, were visualized under UV light. No-template cDNA-negative controls were also included for each primer pair.

### 2.5. Quantitative Real Time PCR (qRT-PCR) Assay

Based upon whether OR genes were PCR amplified in the RT-PCR assays, a subset of ORs were selected for qRT-PCR analysis in order to assess the effect of mating status on OR expression in the *C. pomonella* female abdomen tip. qRT-PCR experiments were carried out with a Roche Light Cycler 480 II thermocycler (Roche, Basel, Switzerland), using the primer pair that worked best in the RT-PCR assays (e.g., single amplicon product and brightest band under same amplification settings). For each primer assay, amplification was performed on biological triplicates of both unmated and mated abdomen tip samples, with technical duplicates for each sample. For all reactions, the following contents were added: 2 μL of cDNA sample at 12.5 ng/μL, 12.5 μL of LightCycler 480 SYBR Green 1 Master mix by Roche (Indianapolis, IN, USA), 9.5 μL water, and 0.625 μL of of each gene-specific primer at 10 μM/μL (thus, 250 nM final concentration of each primer), for a final reaction volume of 25 μL. The PCR amplification protocol was as follows—initiation phase: 5 min at 95 °C; amplification phase (40 cycles): 10 s at 95 °C, 10 s at 58 °C, 10 s at 72 °C; melting curve phase: 65–95 °C gradient, with analysis every 0.5 °C. Melting curves were analyzed to verify the specificity of amplification products. Primer efficiencies were calculated for all primer pairs using a cDNA sample pooled from all unmated and mated abdomen tip samples, with five serial dilutions from undiluted cDNA and a dilution factor of 2.0 (i.e., 50 ng, 25 ng, 12.5 ng, 6.25 ng, 3.125 ng). For each OR, relative expression was normalized to expression of one of two reference genes, GAPDH or Actin; these reference genes were selected from a pool of eight candidate reference genes [43], based upon optimal stability of expression across all biological samples and primer efficiency values as close as possible to 100%. Candidate reference gene sequences are identified either in the transcriptome of this report, or else the published *C. pomonella* genome [28]. Relative gene expression differences across unmated and mated samples were assessed by determination of relative quantities (primer efficiency value^ΔCT) of the OR gene of interest relative to the relative quantities of one of the reference genes [44,45]. Relative gene expression was determined as a ratio of the average of the mated biological samples versus the average of the unmated biological samples, for final presentation of the results. Assessment of statistical differences across mated/unmated conditions was made using independent *t* testing with two-tailed distribution and two-sample equal variance; significance was assessed at (*p* < 0.05). For final presentation of the results and statistical analyses, relative gene expression values were first log2 transformed and subsequently plotted in Excel.

### 2.6. Chemosensory Receptor Phylogenetic Analysis

For a comparative assessment of ORs, GRs, and IRs, phylogenetic analyses were performed on chemosensory receptors of each gene family expressed in adult antennal and abdomen-tip tissues of *C. pomonella* [27] (Appendix A) in relation to repertoires from other insect species. For all *C. pomonella* chemosensory receptor protein sequences, the most complete version of the ORF was used in the alignment and phylogenetic build, whether the sequence was derived from our previous report [27], the genome database [28], or the current study. For ORs, comparisons were made to sets from *Bombyx mori* [46,47], *Epiphyas postvitanna* [48], and *S. littoralis* [49]. For GRs, comparisons were made to sets from *B.mori* [50], *Heliconius melpomene* [46], and *Plutella xylostella* [51]. For IRs, comparisons were made to sets from *Acyrthosiphon pisum*, *Apis mellifera*, *B. mori*, *Drosophila melanogaster*, and *Tribolium castaneum* [52]; in an attempt to provide greater resolution of phylogenetic relationships for novel CpomIRs, a secondary IR phylogeny was produced utilizing most insect IR/iGluR sequences from the dataset of Croset et al., 2010 [52] as well as novel divergent IRs recently reported for *S. littoralis* [53], and previously reported IRs from other lepidoptera including *E. postvitanna* [48], *M. sexta* [54,55], and *Danaus plexippus* [56]. Amino acid sequences for each gene family were aligned using MAFFT online version 7.220 (https://mafft.cbrc.jp/alignment/server/, accessed on 12 December 2023) through the FFT-NS-I iterative refinement method, with JTT200 scoring matrix, “leave gappy regions” set, and other default parameters [57]. Aligned sequences were used to build the unrooted phylogenetic tree using PhyML 3.0 (http://www.atgc-montpellier.fr/phyml/, accessed on 12 December 2023) [58] using the BioNJ algorithm and maximum likelihood tree with Smart Model Selection (SMS) method [59] with selection criterion set to the Bayesian Information Criterion. This software tool integrated into the PhyML web server automatically selects the best substitution model. For the ORs and GRs, the JTT+R+F model was selected, while for the IRs, the WAG+R+F model was selected. PhyML uses both NNI (nearest neighbor interchanges) and SPR (subtree pruning and regrafting) methods to rearrange and optimize the tree structure. Clade support for maximum likelihood analysis was assessed using the Shimodiara–Hasegawa approximate likelihood ratio test (SH-aLRT) [60]. The nodes with support values SH-aLRT > 0.9 were considered well supported, nodes with values ranging from 0.8 to 0.9 were considered weakly supported, and node values < 0.8 were considered unsupported [58]. A consensus Newick format tree was visualized and processed in MEGA-X software (version 10.2.2) [61] and the final tree output was edited with Adobe Illustrator (version 27.9).

## 3. Results

### 3.1. Transcriptome Overview

A de novo *C. pomonella* female abdomen-tip transcriptome was assembled to investigate chemosensory receptor expression in the PG-OV complex. The Illumina HiSeq PE150 approach unveiled a total of 87.39 million clean sequenced reads from pupal, unmated adult, and mated adult samples that were assembled with the Trinity assembler. In total, 204,868 transcripts (>201 nt) were assembled and organized into 165,662 transcript clusters, with a mean length of 658 nts, an N50 of 972, and 30,726 sequences greater than 1000 nts. Analysis of the transcriptome with the BUSCO Arthropoda orthologue set resulted in hits for 99.31% of queried sequences, with 97.43% identified as complete, indicating a satisfactory overall completeness of the transcriptome (Appendix A).

### 3.2. Odorant Receptors

In the abdomen-tip transcriptome, transcripts encoding 37 ORs were identified (Appendix A), including 29 previously annotated antennal-expressed ORs and 8 ORs that were not identified and annotated in that report [27] (Table 1). These non-annotated ORs were compared phylogenetically to ORs of closely related species (Figure 1) and named according to homology with ORs of another tortricid moth, *E. postvitanna*, or else named according to the nearest sequentially available OR. Three of these ORs presented as putative paralogues of previously identified ORs (OR11, OR12, and OR37), and were thus named respectively (OR11.2, OR12.2, and OR37.2). With the exception of OR69, all of the newly named ORs clustered with putative general odorant receptors; OR69 grouped within the newly expanded novel PR clade [62]. Of the 29 OR transcripts previously annotated, only 2 contained complete ORFs, and another 2 contained ORFs of at least 80% completeness. All other OR transcripts (33 of 37) presented short incomplete ORF fragments. Notably, a candidate PR that was previously reported to be expressed in female *C. pomonella* abdomen tips, CpomOR1 [35], was not identified in our transcriptome.

On account of the fragmentary nature of most OR transcripts, we used RT-PCR amplification to verify expression of all ORs in adult abdomen-tip tissue; for this, source RNA samples from adult mated and unmated *C. pomonella* were combined. Due to the absence of CpomOR1 in our transcriptome, we expanded our RT-PCR study to target OR1 and eight other candidate PRs, from the canonical [9] and novel [8] PR clades, that were not identified in the transcriptome. Thus, a total of 46 ORs were screened by RT-PCR. Ultimately, 32 ORs including CpomOrco expressed in adult female abdomen-tip tissue were validated with RT-PCR (Appendix A), including 26 of 37 that were identified in the transcriptome and 6 additional candidate PRs that were not detected in the transcriptome.

### 3.3. Gustatory Receptors

Twenty-two annotated GR transcripts were identified in the female abdomen-tip transcriptome (Appendix A), including eight candidate novel GRs not previously described in our previous olfactory transcriptome report [27] nor identified in the *Cydia pomonella* genome database [28] (Table 1). Phylogenetic analysis was conducted for all previously described and newly identified candidate GR proteins in comparison with genomic-derived GR datasets from other lepidopteran insects (Figure 2). Novel candidate GRs were named according to homology with other moth GRs in the phylogeny to the greatest extent possible. All three previously identified candidate carbon dioxide (CO_2_) receptor genes were identified in the abdomen-tip transcriptome, as well as three candidate sugar receptors, including one novel representative, GR11, from the fructose receptor subfamily. The remaining GRs abdomen-tip-expressed GRs, including seven novel candidate GRs (GR13, GR41, GR54, GR56, GR57, GR58.2, and GR59), clustered broadly across various clades of putative bitter-compound receptors. RT-PCR analysis confirmed expression in the adult female abdomen tip of all GRs identified in the transcriptome with the exception of GR30 (Appendix A).

### 3.4. Ionotropic Receptors

Eighteen annotated IR transcripts were identified in the female abdomen-tip transcriptome (Appendix A), including those encoding the IR co-receptors, IR25a and IR76b. Four novel candidate IRs (Table 1), as well as four novel iGluRs (Appendix A), not previously identified in our antennal transcriptome [27], nor identified in the *Cydia pomonella* genome database [28], were found. Based upon phylogenetic analysis of *C. pomonella* IRs compared with IR and iGluR sets from diverse insect taxa [52] (Figure 3, Appendix A), a novel cluster of divergent IRs were identified, IR60a.2–IR60a.5. Similarly, the previously identified CpomIR4 [27] has been re-classified as a candidate IR31a homologue based on current phylogenetic information and has been replaced with a newly described CpomIR4 gene that is similar to other lepidopteran-specific IR4 homologues. RT-PCR analysis confirmed expression in the adult female abdomen tip of all IRs identified in the transcriptome with the exception of IR60a.2, IR76b, and IR4 (Appendix A).

### 3.5. Mating Effect on OR Expression in Adult Female Abdomen Tip

We hypothesized that for some of the ORs, expression might be regulated by mating, and further, ORs for which expression is induced by mating may have special relevance to detection of odorants that influence egg-laying behavior. We thus attempted quantitative real-time PCR (qRT-PCR) assays on all 32 ORs for which expression was validated by RT-PCR assay. Initial testing using multiple sets of primers for each OR revealed that for 18 of these 32 ORs (56%), expression was not consistently detectable with the qRT-PCR assay. We thus pursued fully replicated qRT-PCR assays of the remaining 14 ORs (44%) to determine whether mating affects their expression. It was determined that for one of the ORs, OR70, expression was significantly increased in the abdomen tips of mated versus non-mated female codling moths, and for two other ORs, OR47 and OR63, expression was significantly decreased in the abdomen tips of mated versus non-mated female codling moths (Figure 4). For all other ORs assayed, there was no significant difference for expression in the abdomen tips of mated versus unmated females.

## 4. Discussion

We identified 37 ORs, 22 GRs, and 18 IRs in our female abdomen-tip transcriptome. While BUSCO analysis of the transcriptome indicated a high degree of completeness, a great majority of chemosensory receptor transcripts contained short, incomplete ORF fragments, indicative of low expression coverage. Initial quantitative analyses of OR abundance estimates were generally low (less than 1), and thus were not considered further for analysis and reporting in this study.

Consistent with this, candidate PR CpomOR1 transcripts were not detected in the transcriptome despite previously being detected in female abdomen tips by RT-PCR assay [35] and confirmed in this study (Appendix A). It may be the case that OR1 displays restricted expression in codling moth abdomen tip neurons below thresholds levels for detection in our transcriptome due to possible limitations of transcriptomic methods to identify the expected chemoreceptors in the various sensory organs from lepidopterans [63]. While the ovipositor of *C. pomonella* has not been previously examined directly for chemosensory function, very few chemosensory sensilla and OSN types were identified on the ovipositor of another moth, *M. sexta* [17]. Similar olfactory morphology and physiology in *C. pomonella* would be consistent with low expression in olfactory neurons below transcriptomic detection thresholds.

Based upon a combination of transcriptomic and RT-PCR analyses, transcripts encoding 32 ORs, 21 GRs, and 15 IRs were confirmed to be expressed in the female adult moth abdomen tip, which are among the highest reported counts to date within Lepidoptera. Comparably, in *M. sexta,* expression of 30 ORs, 14 GRs, and 15 IRs was reported specifically for the ovipositor tissue [55], while in the PG-OV of *Helicoverpa assulta*, 22 ORs, 6 GRs, and 13 IRs were identified in a transcriptomic study, and in the PG-OV of *S. frugiperda*, 12 ORs, 4 GRs, and 10 IRs were revealed. We were not able to detect or confirm expression in female adult abdomen tips for 11 of the 37 ORs nor one of the GRs and three of the IRs identified in the abdomen-tip transcriptome. It is possible that some of these are not expressed in the female adult abdomen tip, but rather only in the abdomen tip of the pupal stage, since pupal abdomen tips were also used to generate the source transcriptome. Given the overall objectives of this project, we did not conduct PCR screening of chemosensory receptors in the pupal stage. Alternatively, failure to detect expression by PCR in the female adult abdomen tips may be due to technical reasons, for example, faulty oligonucleotide primers or expression below detection thresholds. To rule these out, for genes whose expression we were unable to detect during initial PCR assays, we designed and tested additional oligonucleotide primers, and moreover, as a positive control for tested primer sets, we PCR assayed expression in adult antennae, as most of the genes characterized in this study were previously observed to be expressed in codling moth antennal tissue [27,64].

Previously, we reported at least 61 ORs expressed in the antennae of adult codling moth [27]; substantial expression overlap was observed, in that 27 of these ORs were also confirmed to be expressed in the female adult codling moth abdomen tip. This overlap is not surprising considering that it has been shown that apple volatiles can influence both attraction and oviposition behaviors in female codling moth [3,65,66]; specifically, the odorant α-farnesene has been shown to induce attraction [67] and stimulate oviposition [68]. Likewise, the codling moth male and female attractant, pear ester ((E,Z)-2,4-ethyl decadienoate) was also observed to stimulate oviposition by gravid codling moth females [69]. Transcripts encoding a pear ester receptor for codling moth, CpomOR3 [31], were detected in our transcriptome in this study, and this receptor was also the most highly expressed OR in female codling moth antennae [27]. Moreover, one of these ORs, CpomOR29 clusters within a conserved sub-family that includes HassOR31 (see Figure 1), which was previously reported to be highly expressed in the ovipositor of *H. assulta*. and functionally characterized as having a role in mediating oviposition behaviors [18].

Conversely, eight ORs (OR11.2, OR12.2, OR23, OR34, OR37.2, OR52, OR69 and OR70) were detected in our abdomen-tip transcriptome that were not previously identified in our adult antennae/larval head transcriptome [27]. With the exception of OR69, all of these are found, phylogenetically, scattered across various non-PR clades; it may be hypothesized that they would function in mediating female behaviors related to oviposition. On the other hand, OR69 clusters within the recently expanded novel PR clade [8,62]. Multiple ORs within this clade display sex-biased antennal expression in codling moth [27] and may play a role in mediating intra-species communication. Including OR69, 8 of 11 codling moth ORs within this clade were identified in our female abdomen-tip transcriptome; however, none of them have yet been functionally characterized.

The detection of candidate PRs in the adult female abdomen tip, from both the canonical lepidopteran PR clade (OR1, OR4, OR5, OR7, OR8, OR21, OR22), as well as the novel expanded PR clade (OR30, OR31, OR40, OR63, OR64, OR69), indicates olfactory communication may occur within species at close range, possibly influencing mating behaviors and also feedback inhibition as it relates to pheromone release or oviposition deterrence. It has been hypothesized that female auto-detection of pheromone at the ovipositor tip could modulate feedback regulation of pheromone release by the pheromone gland [22,35]. It must be noted, however, that at least one OR from the canonical pheromone clade, OR3, has been shown to respond to a host plant kairomone [31]. Given that 14 compounds have been identified in the female codling moth pheromone gland [70] and as many as 22 ORs have been identified in the canonical and novel PR clades in this report and previously [27], it is expected that several more of these may also detect non-pheromone compounds, mediating interspecific interactions related to oviposition or other behaviors.

In as much as it has been shown that host plant volatiles can positively stimulate oviposition in *C. pomonella* [66,69], an olfactory role for oviposition deterrence has also been demonstrated. Avoidance of oviposition or reductions in egg-laying rates have been observed in presence of fatty acid and ester compounds extracted from eggs of another moth, *Lobesia botrana* [71], solvent-extracted odorants from cardamom, *Elettaria cardamomum* [72], non-host plant chemical extracts [73], and the primary codling moth pheromone compound codlemone [74]. The presence of both general odorant and candidate PRs expressed in the female adult codling moth abdomen tip suggests that olfactory function in this region may be guiding oviposition behavior at close range.

Contrary to evidence for olfactory cues either stimulating or deterring oviposition by codling moth females, little is known about the role of gustatory cues mediating contact chemoreception as it relates to oviposition for this species; to our knowledge, no studies to date have examined this phenomenon in *C. pomonella*. Gustatory function has been previously reported for chemosensory neurons found within sensilla on the ovipositor of another moth, *S. littoralis* [19]. Within the family Tortricidae, chemosensory sensilla were identified on the ovipositor of the spruce budworm, *Choristoneura fumiferana*, and gustatory function was attributed to one type of sensilla containing neurons responsive to sucrose and potassium chloride [75].

Contact chemoreception was reported to be essential in mediating oviposition deterrence for the noctuid moth, *S. litura*, in response to the plant metabolite, Rhodojaponin-III [76]. In codling moth, physical features on leaf and fruit, including wax, have been observed to influence oviposition decisions [77]. In other moths, it has been noted that mechanosensory hairs are predominant on the PG-OV complex compared with chemosensory hairs [19,75], and these would influence oviposition decisions related to physical features of plant matter. Conversely, a role for contact chemodetection of plant metabolites in the waxy surface layer of leaf and fruit matter has also been suggested to influence oviposition determination [78]. The presence of 21 GRs confirmed expressed in the adult female codling moth abdomen tip, spanning the CO_2_, sugar, and bitter-compound receptor sub-families is a strong indication that contact chemoreception is a factor in oviposition decisions in *C. pomonella*.

In addition to ORs and GRs, 18 IRs were detected in our female abdomen-tip transcriptome, 15 of which were confirmed to be expressed in adult females. This included six subunits from the antennal subfamily (IR21a, IR31a, IR64a, IR68a, IR75d, IR93a), five lepidoptera-specific antennal IRs (IR1, IR75p.1-3, IR75q.2) [79,80,81], four subunits from the divergent IR subfamilies (IR60a.2-5) and one subunit that has not been classified yet (IR4), as well as two of the IR co-receptors (IR25a and IR76b) [10,52].

Among these IRs, some have been functionally characterized in *D. melanogaster* demonstrating their response to acids. These include IR64a, which was first identified as a possible CO_2_ sensor from neurons of the *D. melanogaster* sacculus innervating DC4 glomeruli and responding to carbonic acid but not to bicarbonate ions, suggesting that these neurons detect acidosis produced by increased CO_2_ concentrations, rather than CO_2_ itself [82]. Other studies demonstrated the expression of IR64a in two subpopulations of neurons: one from the ventral sacculus, innervating DC4 glomeruli, and one from the dorsal sacculus, innervating DP1m glomeruli, and reporting axonal branching from these glomeruli to distinct regions of the lateral horn [83]. While neurons innervating DP1m responded to a wide panel of odorants, neurons innervating DC4 showed a specific complementary response to acids, such as formic acid, HCl, and HNO_3_, to which neurons innervating DP1m gave a lower response amplitude.

Other IRs that we found in the abdomen tip include subunits from the IR75 clade, among which, most are renowned for acid binding, like proteins expressed from the *D. melanogaster* IR75cba locus [84,85,86,87]. In particular, our analysis unveiled the PG-OV expression of IR75d, for which the orthologue from *D. melanogaster* is expressed in ac1, ac2, and ac4 antennal neurons, and contrary to IR75cba subunits [85,86], is renowned for sensing the polyamine pyrrolidine, rather than acids [87]. However, in a recent project, we demonstrated that the IR75d orthologue from *D. suzukii* responds to hexanoic acid [88]. In accordance with a possible role for IR75d as an acid sensor, other research on dipterans unveiled IR75s that are not orthologues of the DmelIR75a, b, and c, tuning primarily to carboxylic acids. These include IR75k1 and IR75k3 from *Aedes aegypti* and IR75e from *Aedes albopictus*, binding carboxylic acids ranging from seven to nine carbons [89], or IR75k from *A. gambiae*, binding carboxylic acids ranging from six to ten carbons [90].

Research on lepidopterans unveiled additional copies of IR75 isoforms that are not orthologous to DmelIR75a, b, and c [52,81,91], such as the Lepidoptera-specific antennal IRs, IR75p and IR75q [80]. In *Agrotis segetum*, IR75p.1 and IR75q.1 respond to medium-chain fatty acids, with hexanoic acid being the most potent agonist for AsegIR75p.1 [91]. In *Conogethes pinicolalis* (Lepidoptera: Crambidae), IR75ps have shown antennal expression with a female (IR75p2) or a male bias (IR75p and IR75ps) suggesting their co-involvement with OR subunits in olfactory-related modalities [92]. Despite these findings on IR75p and IR75q ligand binding, there is no evidence reported from the orthologues of *Bombyx mori*, which are the closest to the *C. pomonella* subunits that we found in the PG-OV (Figure 3).

Other acid-binding IRs include IR31a, forming IR8a-dependent cation channels expressed in *Drosophila* ac1-neurons, responding to 2-oxopentanoic acid and projecting into VL2p and vVL2p glomeruli [87]. Other candidate IRs for binding acid ligands include the Lepidoptera-specific antennal IR1 and the uncharacterized IR4, based on phylogenetic evidence from a recent study on *Odontothrips loti* (Thysanoptera: Thripidae), which reported these subunits clustering with the IR75a clade, with IR1 showing an antennal bias for both sexes and the IR4 subunit showing body bias [93]. Identification of IR1 and IR4 in the PG-OV of *C. pomonella* is in accordance with previous findings reported for *Helicoverpa zea* [26] where homologues have been found co-expressed with other IRs in the same organ of this moth.

Although RNA-sequencing analysis of the *C. pomonella* PG-OV unveiled subunits involved in acid sensing, we did not identify expression of the IR8a co-receptor, which is renowned for forming functional heterotetramers with acid-detecting IRs [10,87,94]. Previous studies demonstrated a complex co-expression [95] and functionality pattern [96] of co-receptors from IRs, ORs, and GRs, which is common in the olfactory neurons of both *D. melanogaster* and other insects. Considering all lines of evidence, we cannot assume that the absence of IR8a may result in a lack of functioning for the subunits renowned for forming acid-sensing cation channels with the latter. It is possible that other co-receptors such as IR25a and IR76b would form alternative cation channels with these IRs. In a parallel scenario, the IRs that we have found expressed may be at the base of physiological processes differing from what have up to now been reported for their chemosensory modalities: IR expression may result in alternative functions in the PG-OV, such as ion/voltage homeostatic processes or thermal and hygrosensations.

Indeed, apart from IR subunits renowned for acid sensing, other IRs that we found in the abdomen tip of *C. pomonella* are renowned for thermal or hygroscopic signal transduction. Among these, IR21a has been found expressed both in the antennal neurons of the arista and the sacculus [10] and in the dorsal organ of larval heads [97]. Findings from one of our recent studies suggest a similar expression and functional pattern in the codling moth [27], where the IR21a receptor was expressed in the heads of neonate larvae. Stimuli for the aristal-expressed IR21a are unknown [87]. However, physiological experiments have found that aristal neurons respond to either increases or decreases in temperature [98]; more recently, other reports suggested IR21a constituting IR25a-dependent sensors in the larval dorsal organ of *D. melanogaster* with a role in thermotransduction-mediated thermotaxis of the insect [99]. Interestingly, another IR subunit that we found expressed in the abdomen tip is IR93a, which is renowned for mediating thermosensation, as well as hygrosensation, while another IR, IR68a, is also required together with IR40a to sense humidity in the moist cells of the *Drosophila* antenna [100,101,102,103,104]. Except for IR40a, all of these subunits are expressed in the *C. pomonella* PG-OV, adding evidence for thermo/hygroscopic sensing to a possible IR-based chemical sensing.

Aside from candidate chemical sensors and thermo/hygroscopic sensors, RNA-seq from the *C. pomonella* PG-OV transcriptome unveiled a cluster of IRs nested within a divergent IR sub-family. Based on our analyses, it was difficult to resolve phylogentic relatedness of these IRs to functionally conserved clades; relationships to some putative IR60a genes were established (Appendix A), so we have tentatively classified them as IR60a.2-IR60a.5, though this may be subject to future revision. In *Drosophila*, divergent IRs are renowned for being expressed in peripheral and internal gustatory neurons; they are expected to have evolved under weaker purifying selection, containing more sites that have been shaped by positive selection, and contrary to antennal IRs, they are not as conserved across the different insect orders [52]. Interestingly, the *C. pomonella* IR putative paralogues IR60a.2–IR60a.5 are among the novel chemosensory receptors that are expressed in the female abdomen-tip transcriptome (Table 1). Future studies may investigate functional evidence of their expression, most probably demonstrating their involvement in taste perception, as expected for IR members of the divergent IR subfamilies [52,105].

Independently from the expected functionalities of the IR subunits that we found expressed in the *C. pomonella* PG-OV, the renowned involvement of IRs in both olfaction and taste and their co-expression with odorant (ORs) and taste (GR) receptors may further accord with a lack of evidence of a precise differentiation between these two senses in this organ [106]. To investigate this hypothesis further, in situ experiments are needed to demonstrate the co-localization of IRs, ORs, and GRs within the same cells. Adjusting protocols described for the identification of PRs from other lepidopteran species [22] may facilitate investigations in this regard.

Finally, we examined effects of mating on OR expression in adult female abdomen-tip tissue using qRT-PCR assay. Due to low expression abundance of OR transcripts in the abdomen tip, we were only able to consistently detect expression of less than half of the ORs assayed (14/32). Among these results, mating effect was restricted to only a few receptors; OR47 and OR63 were observed to be downregulated after mating, while OR70 was observed to be upregulated. This contrasts to *M. sexta,* in which a mating effect on OR expression in the ovipositor was not observed [55]. OR63 is found within the expanded novel PR clade; downregulation in the abdomen tip after mating suggests it may function in mediating mating or other reproductive behaviors unrelated to oviposition. Conversely, OR47 and OR70 are found in different OR sub-families outside of the PR clades and are hypothesized to respond to general odorants related to host-site localization and oviposition, respectively. These ORs will be subjected to further research to investigate their role in codling moth behaviors through assays on OR protein function [30] as well as CRISPR knock-out experiments [35].

## Figures and Tables

**Figure 1 insects-14-00948-f001:**
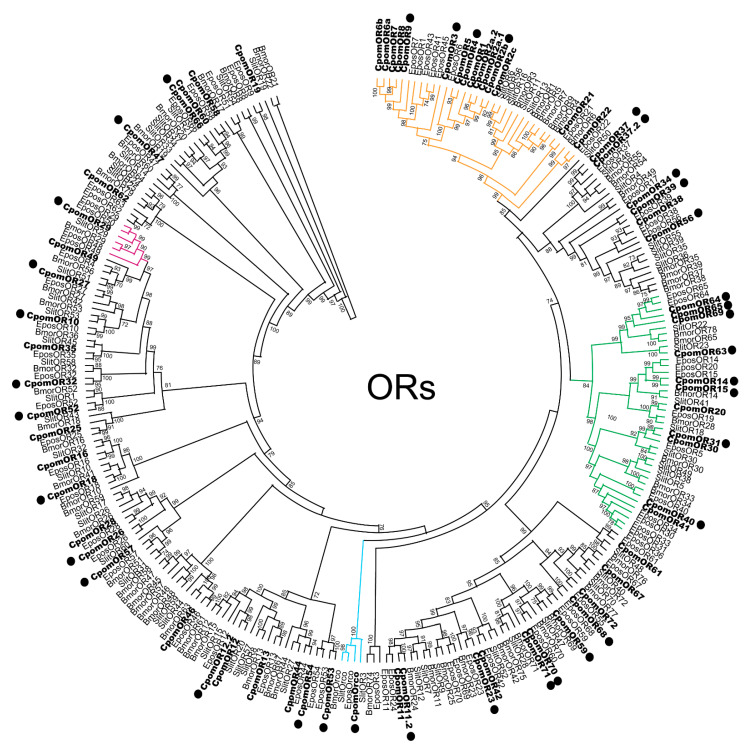
Maximum likelihood phylogenetic tree of candidate CpomOR sequences with other lepidopteran OR sequences. Unrooted phylogenetic tree built using the online tool PhyML 3.0. Includes sequences from *Cydia pomonella* (Cpom), *Epiphyias postvittana* (Epos), *Bombyx mori* (Bmor), and *Spodoptera littoralis* (Slit). Branches of the Orco clade are colored light blue; branches of the lepidopteran canonical “pheromone receptor” clade are colored orange; branches of the expanded novel pheromone receptor clade are colored green; a conserved clade containing an OR known to be expressed in the ovipositor and function in oviposition [18] is colored magenta; *C. pomonella* ORs are indicated with a larger bold font; *C. pomonella* ORs identified in the abdomen-tip transcriptome are marked with a “•”. Node support was assessed with the Shimodiara–Hasegawa approximate likelihood ratio test (SH-aLRT); values greater than 0.7 are shown.

**Figure 2 insects-14-00948-f002:**
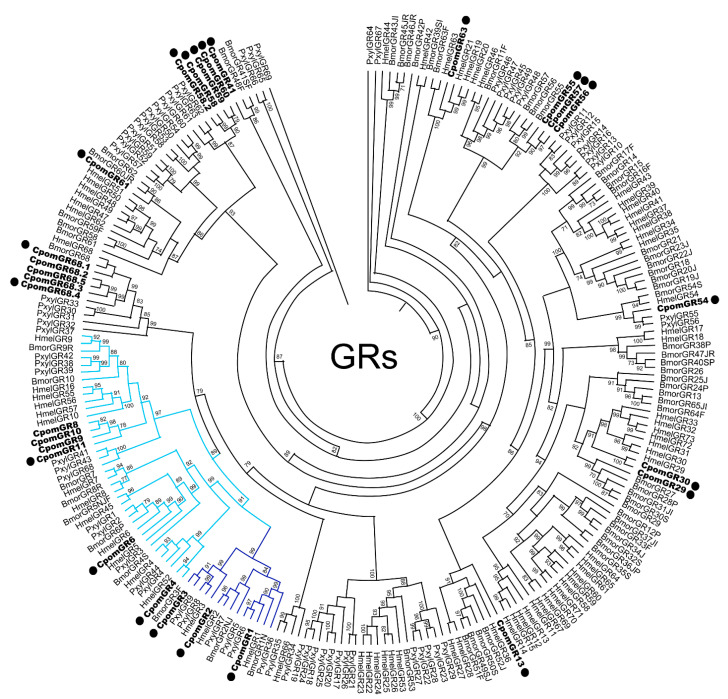
Maximum likelihood phylogenetic tree of candidate CpomGR sequences with other lepidopteran OR sequences. Unrooted phylogenetic tree built using the online tool PhyML 3.0. Includes sequences from *Cydia pomonella* (Cpom), *Bombyx mori* (Bmor), *Heliconius Melpomene* (Hmel), and *Plutella xylostella* (Pxyl). Branches containing putative carbon dioxide receptors are colored dark blue; branches containing putative sugar receptors are colored light blue; branches containing putative bitter receptors are colored black; *C. pomonella* GRs are indicated with a larger bold font; *C. pomonella* GRs identified in the abdomen-tip transcriptome are marked with a “•”. Node support was assessed with the Shimodiara–Hasegawa-approximate likelihood ratio test (SH-aLRT); values greater than 0.7 are shown.

**Figure 3 insects-14-00948-f003:**
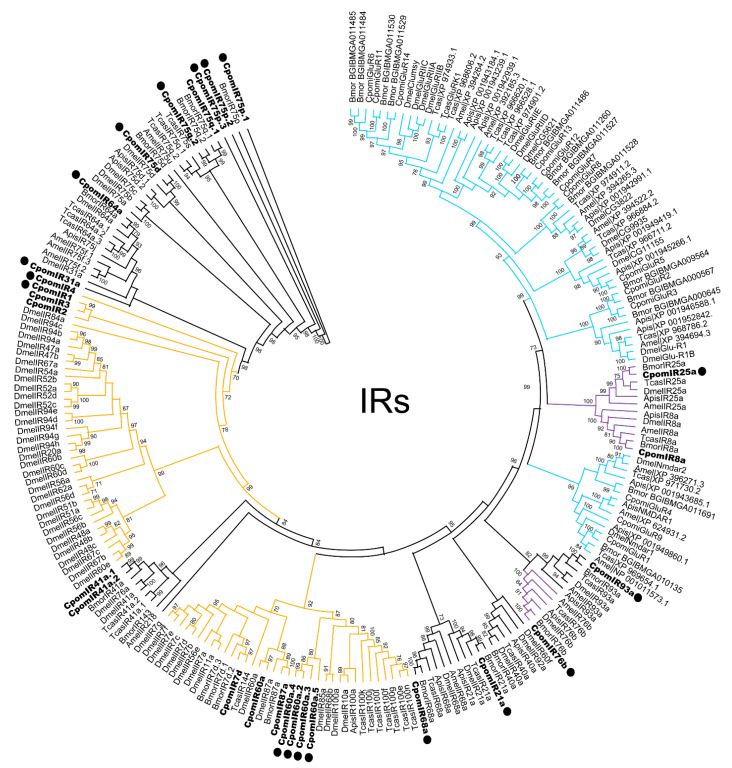
Maximum likelihood phylogenetic tree of candidate CpomIR/iGluR sequences with other insect IR/iGluR sequences. Unrooted phylogenetic tree built using the online tool PhyML 3.0. Includes sequences from *Cydia pomonella* (Cpom), *Bombyx mori* (Bmor), *Acyrthosiphon pisum* (Apis), *Apis mellifera* (Amel), *Drosophila melanogaster* (Dmel), and *Tribolium castaneum* (Tcas). Branches containing putative ionotropic glutamate receptors (iGluRs) are colored light blue; branches containing putative IR co-receptors are colored purple; branches containing divergent IRs are colored orange; branches containing putative antennal IRs are colored black; *C. pomonella* IRs are indicated with a larger bold font; *C. pomonella* IRs identified in the abdomen-tip transcriptome are marked with a “•”. Node support was assessed with the Shimodiara–Hasegawa approximate likelihood ratio test (SH-aLRT); values greater than 0.7 are shown.

**Figure 4 insects-14-00948-f004:**
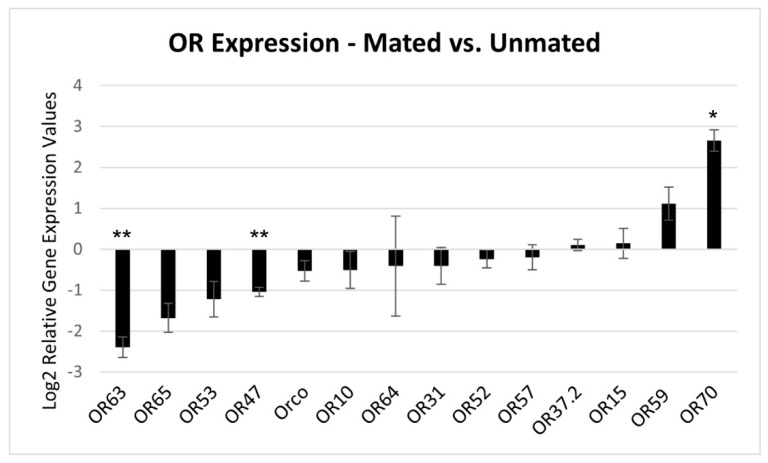
Effect of mating on expression of ORs in adult female abdomen-tip tissue. Binary log-transformed relative gene expression values shown, normalized to reference gene (either GAPDH or Actin), with average of three biological replicates from mated moths each calibrated to average of three biological replicates from unmated moths. For each bar plot, values less than zero are indicative of reduced expression after mating, while values above zero are indicative of increased expression after mating. Statistical assessments of differences in values between log2 transformed mated and unmated values conducted with Student’s *t* test (two-tailed distribution with two-sample equal variance); “*” indicates *p*-value less than 0.05; “**” indicates *p*-value less than or equal to 0.005. Error bars are standard error values.

**Table 1 insects-14-00948-t001:** Summary of novel chemosensory receptors expressed in female abdomen-tip transcriptome.

Name	ORF Length	ORF Completion Status	Best PSI-Blast Hit to Chemosensory Receptor in NCBI-nr Database	E-Value
CpomGR11	415	Complete	*L. glycinivorella* GR68-like	9 × 10^−29^
CpomGR13	131	Incomplete	*L. glycinivorella* GR43a-like	3 × 10^−30^
CpomGR41	144	Incomplete	*L. glycinivorella* uncharacterized protein (7tm_7 chemosensory receptor)	1 × 10^−61^
CpomGR54	129	Incomplete	*Operophtera brumata* GR30	1 × 10^−39^
CpomGR56	147	Incomplete	*Eogystia hippophaecolus* GR1	1 × 10^−10^
CpomGR57	116	Incomplete	*E. hippophaecolus* GR1	7 × 10^−11^
CpomGR58.2	219	Incomplete	*H. nubiferana* GR5	2 × 10^−8^
CpomGR59	223	Incomplete	*Hedya nubiferana* GR5	6 × 10^−29^
CpomIR60a.2	343	Incomplete	*Heliconius erato petiverana* IR60a1a	2 × 10^−29^
CpomIR60a.3	310	Incomplete	*Achelura yunnanensis* IR100a	2 ×10^−36^
CpomIR60a.4	563	Incomplete	*Galleria mellonella* IR21a-like	9 × 10^−53^
CpomIR60a.5	571	Incomplete	*Spodoptera litura* IR100p	3 × 10^−42^

## Data Availability

All data generated and analyzed in this study are included in the published article and its Appendix A. Transcriptome raw reads sequence data are available through the NCBI Sequence Read Archive (SRA), (BioProject: PRJNA1020188, SRA Accession Numbers: SRR26151771—SRR26151773 (three samples).

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
