# Peer review of "Chemosensory Receptor Expression in the Abdomen Tip of the Female Codling Moth, Cydia pomonella L. (Lepidoptera: Tortricidae)"

_insects, 2023, doi:10.3390/insects14120948_

Round 1

Reviewer 1 Report

Comments and Suggestions for Authors

The present article reports the repertoire of chemoreceptors as part of the abdomen tip of Cydia pomonella, a well studied Tortricid. From a chemosensory perspective this study is a nice contribution to the field, supporting that chemosensory genes are not solely part of antennae, but also of other morphological structures. Overall, the article is well presented and easy to follow. However, there are some minor revisions that could improve the quality of the work before considered for publication. And my suggestions for that are detailed in the following comments:
L93: ORs instead of olfactory receptors
L97-98: PRs instead of pheromone receptors. Also, I would relate functions mentioned here to the actual chemoreceptor for general readers.
L115: Use abbreviation for pheromone receptors.
L120-123: I believe it is important to give context of Orco’s function, then the example.
L124-126: I think the purpose of this sentence is to address the issue of chemoreceptors in abdomen tip of C. pomonella. However, I feel it can be written better. I suggest to start like in the abstract, saying that most of the research has been focused on antennae, and the role of males. But a study based on CRISPR and the function of OR1 found that…..

L137-138: I think the present methodologies will not fully answer the hypothesis. Instead, findings will allow the identification of a repertoire of chemoreceptors with physiological plasticity that resembles chemosensory functions proper of mating and/or oviposition. Anyways, the idea is that functions at in vivo level would need behavioral assays with manipulation of chemoreceptors.
L233: I think authors should be careful here, as several primer pairs for RT-PCR appeared unspecific in the gel images, and using them for qPCR could have affected calculations.
L250-252: there is a lack of description around how mated and unmated moths were obtained. There is an important aspect of olfaction plasticity where age, starving, mating, odorant exposure, affects the expression of chemosensory genes. So, how authors controlled mating status in females? Where all females the same age when mated and unmated? How they were sure females were mated? These and other details should be included.
L339: I suggest to re-root the tree based on Orco lineage.
L342: I suggest also to highlight the small conserved clade for plant volatiles ORs, where SlitOR14 and BmorOr60 are. It seems not CpomORs are present there, but it gives another point of view for analysis in the context of oviposition behaviors. See Guo et al 2021 paper doi:10.1093/molbev/msaa300
L348: are authors sure that CpomOR11.2 and 69 are novel? Or may these receptors be part of the same sequence reported in column 4? Particularly 11.2 vs Or11 as complete ORF and good e-value.
L348: different letter types in Table 1. 
L356-360. I got lost here. Were 32 or 26 the OR genes amplified and confirmed by RT-PCR?
L377: I think it would be highly informative to highlight those GRs that have been functionally studied, something that could also be applied on Or tree. For such purpose, several Drosophila GRs and their functions have been reported. 
L416: So, 32 were validated, not 26?
L427: this is an uncommon way to show a two-group based plot. In that sense, I understand that mated individuals are above the x-axis, while below are unmated ones? I would suggest to plot again these results as two -group bars for each OR, indicating clearly where mated and unmated samples are.
L442-448: would be worth mentioning how deep (million reads) RNaseq was? 
L454: There are several numbers reported, 37, then 32, or 26, now 33. I understand that dealing with many transcripts and subsequent filters by experiments, ORFs, annotations, led to confusion. Please, give the right number
L457: Helicoverpa assulta
L459-461: that is 27 ORs, please, check the number Ors.
L473: so, 28 in both antennae and adult female abdomen tip. According to Table 1, 8 ORs are reported as novel in abdomen tip. That gives 36 out of 37 initially identified from transcriptome. 
L480-482: However, OR3 was not amplified by qPCR. Could authors give an explanation for this in terms of the function that OR3 could give in the context of mating status?
L512: if this compound was named before, it is not necessary to include the iupac name. Nevertheless, both isomer letters must go within parenthesis.
L525: I guess Spodoptera was already abbreviated before. Should be S. litura.
L535: C. pomonella
L548-552: please, give full name for these abbreviations.
L638: PRs

Author Response

Reviewer 1

The present article reports the repertoire of chemoreceptors as part of the abdomen tip of Cydia pomonella, a well studied Tortricid. From a chemosensory perspective this study is a nice contribution to the field, supporting that chemosensory genes are not solely part of antennae, but also of other morphological structures. Overall, the article is well presented and easy to follow. However, there are some minor revisions that could improve the quality of the work before considered for publication. And my suggestions for that are detailed in the following comments:

L93: ORs instead of olfactory receptors

The change has been made.

L97-98: PRs instead of pheromone receptors. Also, I would relate functions mentioned here to the actual chemoreceptor for general readers.

The change has been made, and a general statement about Lepidopteran pheromone receptors has been made prior to this at an appropriate location in the introduction (L72-73).

L115: Use abbreviation for pheromone receptors.

The change has been made.

L120-123: I believe it is important to give context of Orco’s function, then the example.

The context has been provided at first mention of Orco (L94-95)

L124-126: I think the purpose of this sentence is to address the issue of chemoreceptors in abdomen tip of C. pomonella. However, I feel it can be written better. I suggest to start like in the abstract, saying that most of the research has been focused on antennae, and the role of males. But a study based on CRISPR and the function of OR1 found that…..

The language of this statement has been revised as suggested.

L137-138: I think the present methodologies will not fully answer the hypothesis. Instead, findings will allow the identification of a repertoire of chemoreceptors with physiological plasticity that resembles chemosensory functions proper of mating and/or oviposition. Anyways, the idea is that functions at in vivo level would need behavioral assays with manipulation of chemoreceptors.

That is a fair point. Language concerning the hypothesis has been revised.

L233: I think authors should be careful here, as several primer pairs for RT-PCR appeared unspecific in the gel images, and using them for qPCR could have affected calculations.

This is a good point. To make it clearer, all PCR primers that were subsequently used for the qPCR assays have been marked with underline in Data S5 and the appropriate legends have been updated to reflect this. In the process of doing this, a couple of errors were noted in Figure 4, as such an updated figure 4 has been included in the revised manuscript.

L250-252: there is a lack of description around how mated and unmated moths were obtained. There is an important aspect of olfaction plasticity where age, starving, mating, odorant exposure, affects the expression of chemosensory genes. So, how authors controlled mating status in females? Where all females the same age when mated and unmated? How they were sure females were mated? These and other details should be included.

Thank you for allowing us to clarify. The text describing the mating conditions is already included elsewhere in the methods and is described in detail in section “2.1. Insects and Sample Collection”. For greater clarity, it is indicated on L157-158, that the experimental design applied to all conditions, “with reference to sampling for both RNA-sequencing and qRT-PCR assay”

L339: I suggest to re-root the tree based on Orco lineage.

The decision to not root the tree based on Orco was following from our previous studies, as well as those from other researchers in the field, see for example, Walker et al., 2019; Walker et al., 2016; also Poivet et al., 2013. As we were not conducting a deep evolutionary study, per se, but rather aimed to display how chemosensory receptors were generally related to each other, we decided not to bioinformatically impose a root on the tree, but instead highlight functionally characterized clades, including Orco and putative PR sub-families.

L342: I suggest also to highlight the small conserved clade for plant volatiles ORs, where SlitOR14 and BmorOr60 are. It seems not CpomORs are present there, but it gives another point of view for analysis in the context of oviposition behaviors. See Guo et al 2021 paper doi:10.1093/molbev/msaa300

This is a very good point. In general, the individual non-PR clade ORs were not highlighted because in our previous study, de Fouchier et al., 2017, we reported a broad functional conservation of ORs across all Lepidopteran OR sub-families including candidate PR and non-PR clades, so it did not make sense to highlight all of the different clades. The conserved clade that you mentioned, which is the described HarmOR42 sub-family in Guo et al., 2021, seems to be more responsive to floral odors and relevant to feeding rather than oviposition, so we have chosen not to highlight this clade. However, in a different report, Li et al., 2020, eLife, reported the functional characterization of an OR highly expressed in the ovipositor of Helicoverpa assaulta, HassOR31. This OR is also part of a highly conserved clade that included the abdomen tip expressed OR, CpomOR29. Thus, we have highlighted this clade in the figure and given it mention in the relevant section of the discussion, L497-500

L348: are authors sure that CpomOR11.2 and 69 are novel? Or may these receptors be part of the same sequence reported in column 4? Particularly 11.2 vs Or11 as complete ORF and good e-value.

Please allow us to clarify. Based on this comment and that of another reviewer, it is acknowledged that none of the ORs identified in this study may be novel, when compared to the full set of ORs reported by Wan et al., 2019 in their genomic analysis. We have thus revised any references to novel sequences and re-analyzed our blast results with reference to the C. pomonella genome database (http://v2.insect-genome.com/) instead of the NCBI blast server, and have thus updated Table 1 and Data S4 file to reflect this. While none of the ORs can correctly be referred to as novel, several each of the GRs and IRs were not identified in the C. pomonella genome database. These remain in Table 1, and Data S4 file has been updated to indicate that these were not identified in the genome database.

L348: different letter types in Table 1. 

This has been corrected.

L356-360. I got lost here. Were 32 or 26 the OR genes amplified and confirmed by RT-PCR?

Our apologies for the confusion, the text has been revised to make the point clearer, L367-371. The correct number is 32 ORs that were detected by RT-PCR, including 26 that were detected in the transcriptome and 6 candidate PRs that were not detected in the transcriptome.

L377: I think it would be highly informative to highlight those GRs that have been functionally studied, something that could also be applied on Or tree. For such purpose, several Drosophila GRs and their functions have been reported. 

While the candidate sugar and CO2 receptor GRs are highly conserved across insect orders, there is not well supported clear orthology among the candidate bitter receptors across Diptera (i.e. Drosophila) and Lepidoptera (C. pomonella); thus we did not view it as meaningful to discuss the functional studies on Drosophila bitter-clade GRs. In general, very little functional data is available on lepidopteran bitter-clade GRs, however, we did include text about abdomen-tip expressed candidate sugar and CO2 receptors.

L416: So, 32 were validated, not 26?

Correct, the text from L367-371 has been revised to make the original point clearer.

L427: this is an uncommon way to show a two-group based plot. In that sense, I understand that mated individuals are above the x-axis, while below are unmated ones? I would suggest to plot again these results as two -group bars for each OR, indicating clearly where mated and unmated samples are.

We apologize for the confusion. The title of the figure legend indicates that the figure is showing mating effect on gene expression. And further in the legend it is stated that expression values from the mated moths are calibrated to expression values of the unmated moths. This is a fairly normal way to indicate when results of one condition are normalized to results of another condition. Given that binary log values are shown, a calibrated value of zero would indicate no difference in expression in mated versus unmated. To make this point clearer, text has been added to the figure legend, L443-445, to indicate that values below zero are indicative of reduced expression while values above zero are indicative of increased expression after mating.

L442-448: would be worth mentioning how deep (million reads) RNaseq was? 

The sequencing depth is indicated in the results section on Line 323.

L454: There are several numbers reported, 37, then 32, or 26, now 33. I understand that dealing with many transcripts and subsequent filters by experiments, ORFs, annotations, led to confusion. Please, give the right number

Apologies again for the confusion.

The number 37 includes unique OR gene transcripts identified in the transcriptome, which was derived from pupal and adult abdomen tip samples. All of these were tested to confirm expression in adult abdomen tip samples.

Due to the absence among these 37 ORs of one OR, CpomOR1, that we previously reported to be expressed in abdomen tip in one of our previous studies (Garczynski et al, 2017, J. Eco. Ent.), we conducted an RT-PCR analysis on OR1 and all other candidate PRs, even if they weren’t identified in the abdomen tip transcriptome (n=9 in total), thus testing in total the expression of 46 ORs. Based upon a combination of transcriptomic and RT-PCR analyses, out of these 46 ORs, 32 ORs, including CpomOrco were validated with RT-PCR as expressed in adult female abdomen tip tissue (Figure S1, Data S5). Among these 32 ORs, 26 ORs were identified in the transcriptome. The other 6 ORs are instead candidate PRs that we found expressed by RT-PCR but were not detected in the transcriptome. We hope that our revisions to the original statement (L367-371) have made this all clearer.

L457: Helicoverpa assulta

The typo has been corrected.

L459-461: that is 27 ORs, please, check the number Ors.

That should be 37 ORs, as written, referring to the number of transcripts identified in the transcriptome. Our apologies for the confusion, we hope this has been clarified by our new text added at L367-371.

L473: so, 28 in both antennae and adult female abdomen tip. According to Table 1, 8 ORs are reported as novel in abdomen tip. That gives 36 out of 37 initially identified from transcriptome. 

Correct, though the correct number is 27 not 28, according to Data S4 file. The number has been updated at L491. However, it is clarified here at this point “confirmed to be expressed in adult abdomen tip”, and this confirmation has been provided by RT-PCR assay. Table 1 previously referred to those 8 ORs detected in our abdomen tip transcriptome that were not previously detected in our antennal transcriptome. However only 5 of those 8 were confirmed expressed in adult abdomen tip by RT-PCR (Figure S1/Data S4), thus giving us the total confirmed in adult abdomen tip at 32 ORs (27+5). Regardless, we are no longer calling these ORs novel, since they were reported in the genome publication, and they have been removed from Table 1.

L480-482: However, OR3 was not amplified by qPCR. Could authors give an explanation for this in terms of the function that OR3 could give in the context of mating status?

We revisited RT-PCR detection of OR3, due to our OR3 primers also not initially detecting OR3 in the antennae; thus we redesigned the assay and were finally able to detect OR3 by RT-PCR assay, in the antennae but not the abdomen tip. Appropriate data has been added to the Data S5 file. Failure to detect OR3 by qPCR, however is not unique to OR3, as we were unable to detect a majority of the ORs by qRT-PCR despite being able to amplify them by end-point RT-PCR. This is likely due to general low expression of ORs in the abdomen tip, and this point is already mentioned in the manuscript in the discussion.

L512: if this compound was named before, it is not necessary to include the iupac name. Nevertheless, both isomer letters must go within parenthesis.

Thank you for the clarification. The IUPAC nomenclature has been deleted at this instance, as it is already referred to previously in this manuscript.

L525: I guess Spodoptera was already abbreviated before. Should be S. litura.

Correct. This and other instances of Spodoptera have been abbreviated.

L535: C. pomonella

This and other instances of Cydia have been abbreviated.

L548-552: please, give full name for these abbreviations.

We understand this request, since indicating acronyms of the Drosophila antennal lobe glomeruli may raise questions about origins of their names. However, it is nowadays generally accepted when it comes to mentioning specific glomeruli of the Drosophila AL to just indicate acronyms as they are without mentioning the full name of the glomeruli, this is in accordance with the papers we cited (Ai et al. 2011; Ai et al. 2013).

L638: PRs

This and all other instances of “pheromone receptor” has been abbreviated.

Reviewer 2 Report

Comments and Suggestions for Authors

In this article, the authors characterized the chemosensory receptors that are expressed in the abdomen tip of the codling moth C. pomonella. They conducted RNA-seq and a de novo transcriptome assembly to identify the OR, GR and IRs that are expressed in the abdomen tip from female pupal, adult mated and unmated, and validated the expression using qRT-PCR. Overall, the authors report some genes that are expressed in the abdomen tip and might play an important role in oviposition-site selection, as well as some ORs that are differentially expressed before and after mating. The results discussed in this manuscript are interesting, and I would recommend to accept this manuscript after some revisions.  

The methods and discussion are overall sound, but one major issue for RNA-seq based de novo assemblies is the fragmentation in the assembled transcripts, as the authors report for the chemosensory receptors (Table 1). There is a chromosome level genome available for C. pomonella in which the chemosensory receptors are already annotated, as the authors report in their introduction. The author used these annotations to extract the most complete sequence (line 220), but there is an unclear association of their transcripts and the genome annotations (for instance in Table 1, only 2 ORs correspond to a gene annotated in the genome?; and they report them as novel comparing their report on the antenna expressed receptors, but what about the genome annotation?). This makes me wonder why the authors did not use directly the genome to align the RNA-seq and extract the expressed receptor genes. Therefore, I think that it would be really valuable to align the RNA-seq directly with the genome, and validate the expression of the chemoreceptors based on the quantification of each gene directly from these alignments with the genome annotated genes. Also, the authors should give similar names to the genes from the transcriptome and genome just for reproducibility and not mixing gene names (i.e.: CpomOR69 in Table 1).

In addition, the authors “only” use the receptor genes identified in the antenna and abdomen tip in the phylogenetic trees that were generated using transcriptome de novo assembly (lines 283-286), many of which have incomplete orfs. I would suggest that the authors use here all receptor genes annotated in the genome, as these would have the complete sequences, and the analyses would characterize the complete chemoreceptor repertory in this species and then highlight those genes expressed in the abdomen tip or antenna, which would a valuable figure for the manuscript.   

One minor issue, lines 63 to 70 lack reference.

Author Response

Reviewer 2

In this article, the authors characterized the chemosensory receptors that are expressed in the abdomen tip of the codling moth C. pomonella. They conducted RNA-seq and a de novo transcriptome assembly to identify the OR, GR and IRs that are expressed in the abdomen tip from female pupal, adult mated and unmated, and validated the expression using qRT-PCR. Overall, the authors report some genes that are expressed in the abdomen tip and might play an important role in oviposition-site selection, as well as some ORs that are differentially expressed before and after mating. The results discussed in this manuscript are interesting, and I would recommend to accept this manuscript after some revisions.  

The methods and discussion are overall sound, but one major issue for RNA-seq based de novo assemblies is the fragmentation in the assembled transcripts, as the authors report for the chemosensory receptors (Table 1). There is a chromosome level genome available for C. pomonella in which the chemosensory receptors are already annotated, as the authors report in their introduction. The author used these annotations to extract the most complete sequence (line 220), but there is an unclear association of their transcripts and the genome annotations (for instance in Table 1, only 2 ORs correspond to a gene annotated in the genome?; and they report them as novel comparing their report on the antenna expressed receptors, but what about the genome annotation?). This makes me wonder why the authors did not use directly the genome to align the RNA-seq and extract the expressed receptor genes. Therefore, I think that it would be really valuable to align the RNA-seq directly with the genome, and validate the expression of the chemoreceptors based on the quantification of each gene directly from these alignments with the genome annotated genes. Also, the authors should give similar names to the genes from the transcriptome and genome just for reproducibility and not mixing gene names (i.e.: CpomOR69 in Table 1).

The reason why the RNA-sequencing reads were not aligned to the genome was that this study was initiated, and all transcriptomic analyses conducted herein, prior to the publication of the sequenced C. pomonella genome. It is our opinion that due to generally low expression of the chemosensory receptor genes, that mapping the reads further to the genome will not yield additional useful data. This observation is supported by the fact that we were unable to detect most of the ORs by qRT-PCR (attempted in most cases with multiple primer pairs that did detect expression by end-point RT-PCR), and also that our BUSCO analysis indicated a rather complete transcriptome. The objective of this study was not to provide a quantitative analysis of chemosensory receptor gene expression abdomen tip, but rather to demonstrate broad expression profiles of chemosensory receptor genes as has recently been shown for several other species in the literature.

That said, we do agree regarding the comment about giving similar names from the transcriptome and genome. And we have taken measures to correct the inconsistences and have revised Table 1 and Data S4. However, a couple of caveats must be mentioned with regards to this. First is that in their published report on C. pomonella genome, Wan et al., (2019) only really annotated the ORs among the chemosensory receptors, such as OR3a, OR22a, OR64a, found in their Supplementary Table 13. No such similar names can be found for the GRs nor IRs, neither in the publication/supplemental materials, nor on the genome database (http://v2.insect-genome.com/), so there will not be consistent names provided for these gene families. Furthermore, we have observed that not every gene transcript that we have identified in our transcriptome has a corresponding identified gene in the genome database, especially for GRs and IRs; some genes are simply not found when we blast our transcripts/proteins against the C. pomonella genome database. The best we can do at this point is refer to the generic gene name that is given for those GRs and IRs that are found in the genome database, such as cpo156980. For the ORs, since the nomenclature provided by us previously (Walker et al., 2016) was published prior to the publication of the C. pomonella genome, we have included both names in the Data S4 file to preclude further confusion to the reader.

In addition, the authors “only” use the receptor genes identified in the antenna and abdomen tip in the phylogenetic trees that were generated using transcriptome de novo assembly (lines 283-286), many of which have incomplete orfs. I would suggest that the authors use here all receptor genes annotated in the genome, as these would have the complete sequences, and the analyses would characterize the complete chemoreceptor repertory in this species and then highlight those genes expressed in the abdomen tip or antenna, which would a valuable figure for the manuscript.   

This is partially incorrect, in that for the phylogenetic trees that we did build, we used the most complete version of the protein, whether that was found in our previously published antennal transcriptome, in the abdomen tip transcriptome introduced in this report, or else in the genome. New text has been added to reflect this, L293-296. When the most complete version was found in the genome, we have indicated as much in Data S4, however, not all genes identified in our transcriptome were found in the genome database, and those that were found in the genome database do not always contain complete open reading frames. We have aimed to include the most complete representation of the genes/proteins in our phylogenetic analyses. Data File S4 has been updated to indicate which genes were not found in the genome database.

Furthermore, due to the fact that not all transcripts we identified have representative genes found in the genome database, and additionally that the GRs and IRs from the genome have not been publicly annotated, we argue against expanding further on our current phylogenetic analyses. The aim was to identify which genes are expressed in the abdomen tip transcriptome within the larger gene families, and that has been accomplished. It is clear, for example, whether the abdomen tip expressed ORs cluster with candidate PRs or general ORs; whether the expressed GRs cluster within the conserved candidate sugar/CO2 or else the bitter clades; whether the IRs cluster within the antennal or divergent IR clades. Given a broad lack of functional data, especially on lepidopteran GRs and IRs, we do not think adding in further genes will yield additional further insights.

One minor issue, lines 63 to 70 lack reference.

That is a fair point. An appropriate review article on this topic has been referenced at L70.